# Differences in the Sub-Metatarsal Fat Pad Atrophy Symptoms between Patients with Metatarsal Head Resection and Those without Metatarsal Head Resection: A Cross-Sectional Study

**DOI:** 10.3390/jcm9030794

**Published:** 2020-03-14

**Authors:** Raúl Juan Molines-Barroso, Yolanda García-Álvarez, José Luis García-Klepzig, Esther García-Morales, Francisco Javier Álvaro-Afonso, José Luis Lázaro-Martínez

**Affiliations:** 1Diabetic Foot Unit, Medical Faculty, Complutense University of Madrid, IdISSC, 28040 Madrid, Spaineagarcia@ucm.es (E.G.-M.); alvaro@ucm.es (F.J.Á.-A.); diabetes@ucm.es (J.L.L.-M.); 2Internal Medicine Department, San Carlos Clinic Hospital, IdISSC, 28040 Madrid, Spain; josgar08@ucm.es

**Keywords:** fat pad atrophy, metatarsal head resection, ultrasound, and history of ulceration

## Abstract

We aimed to evaluate the differences in the sub-metatarsal skin and fat pad atrophy between patients at a high risk of ulceration with and without previous metatarsal head resection. A cross-sectional study was performed in a diabetic foot unit involving 19 participants with a history of metatarsal head resection (experimental group) and 19 (control group) without a history of metatarsal head resection but with an ulcer in other locations in the metatarsal head. No participants had active ulcerations at study inclusion. Sub-metatarsal skin thickness and fat pad thickness in the first and second metatarsals were evaluated by an ultrasound transducer. The experimental group showed sub-metatarsal fat pad atrophy (3.74 ± 1.18 mm and 2.52 ± 1.04 mm for first and second metatarsal, respectively) compared with the control group (5.44 ± 1.12 mm and 4.73 ± 1.59 mm) (*p* < 0.001, confidence interval: (CI): 0.943–2.457 and *p* < 0.001, CI: 1.143–3.270 for first and second metatarsal, respectively); however, sub-metatarsal skin thickness was not different between groups (experimental 2.47 ± 0.47 mm vs. control 2.80 ± 0.58 mm (*p* = 0.063, CI: −0.019–0.672) and 2.24 ± 0.60 mm vs. 2.62 ± 0.50 mm (*p* = 0.066, CI: −0.027–0.786) for first and second metatarsal, respectively). Patients with previous metatarsal head resection showed sub-metatarsal fat pad atrophy, which could be associated with the risk of reulceration in the metatarsal head.

## 1. Introduction

Diabetic foot ulcers are considered to be a global health problem [1], with a lifetime incidence of 19–34% [2]. In patients with an ulcer, the 5-year mortality risk is twofold higher than in those without an ulcer [3]. Low healing rates and lower limb amputations are the common factors affecting patients’ quality of life and healthcare costs [4,5]. Although more than 75% of the ulcers can heal after proper treatment, almost half of patients experience a recurrence within 1 year after ulcer healing [2].

Ulcers commonly appear on the plantar surface of the foot [6]. The following risk factors can lead to the development of ulcers: peripheral neuropathy, previous ulcers, foot deformity, and increase plantar pressure [7,8]. Furthermore, the lack of a resolution to these factors can increase the frequency of ulcer recurrence in the plantar surface of the foot [2].

The differences in the mechanical properties of soft tissues are a prognostic factor of diabetic ulcers [9]. Plantar skin and fat pads are the first biologic barriers that protect the plantar surface of the foot from repetitive mechanical stress, absorbing plantar pressure. Any pathologic change in the thickness of these structures reduces the natural protection and increases the risk of ulcers with peripheral neuropathy [10,11].

Unfortunately, the literature data on the predisposing factors of soft tissue atrophy remain inconsistent [12]. In several studies, diabetes mellitus [10] or previous ulceration [13] has been associated with fat pad atrophy; in other investigations, no significant differences were observed between patients with diabetes and those without diabetes [14] with or without previous ulceration [15]. We believe that soft tissue atrophy could be an unresolved risk factor of reulceration, although the risk would be determined by the location of the ulcer.

On the contrary, metatarsal head resection is a surgical procedure commonly performed in patients with metatarsal ulceration. This procedure can eliminate the deformity and heal ulcers affected by osteomyelitis [16]. However, head resection leads to changes in foot biomechanics, which contributes to transfer ulceration in adjacent metatarsal heads [17], suggesting the failure of plantar soft tissue to act as a first line of defense.

The role of plantar skin and fat pads in patients who underwent metatarsal head resection remains unknown. Moreover, no previous studies have provided a comparison between patients with previous metatarsal head resection and those with the same diabetes neuropathy symptoms and previous ulceration occurring in a different location in the metatarsal head.

We hypothesized that sub-metatarsal skin and fat pad thickness were reduced in patients with previous metatarsal head resections compared to patients who had ulcers in locations other than the metatarsal head.

Our study aimed to evaluate the differences in the thickness of the sub-metatarsal skin and fat pads between patients with a history of metatarsal head resection and those without a history of metatarsal head resection.

## 2. Experimental Section

A cross-sectional study was performed between February 2019 and July 2019 involving 38 patients. Patients were admitted to a diabetic foot unit for conservative or surgical treatment for the ulcers. Patients were included in the study during the follow-up period after ulcer healing [18]. Patients aged >18 years, diagnosed with type 1 or type 2 diabetes mellitus according to the criteria of the American Diabetes Association, with peripheral neuropathy and a history of healed ulcers, were included in the study.

The participants were divided into two groups: those (19 patients) with previous ulceration in the metatarsal head that required a metatarsal head resection and those (19 controls) with previous ulceration located in a different region in the metatarsal head.

Patients with active ulcers, previous ulcers caused by trauma, a history of both first and second metatarsal head resection, first or second claw toe deformity, acute or chronic Charcot neuroarthropathy, a history of rheumatoid arthritis, or diseases causing peripheral neuropathy other than diabetes were excluded.

This study was approved by a local ethics committee (approval code no.: 19/077-E; data: 2019/02/13; Hospital Clínico San Carlos, Madrid, Spain) and was performed in accordance with the principles of the Helsinki Declaration. All patients signed a written informed consent form.

Peripheral arterial disease (PAD) was considered when both pedal pulses were absent and/or when the ankle brachial index (ABI) was <0.9. In patients with an ABI of >0.9, a toe pressure of <55 mmHg or a toe brachial index of <0.7 was used to diagnose PAD [19].

Previous ulcers were classified according to the SINBAD (Site, Ischemia, Neuropathy, Bacterial Infection and Depth) classification system [20]. The SINBAD system uses 6 items (ulcer site, ischemia, neuropathy, bacterial infection, area, and depth), scored with 0 or 1 point, to create a SINBAD score of 0–6. Data on ulcers were obtained from the patients’ medical records retrospectively.

Peripheral neuropathy was diagnosed using a 10g Semmes–Weinstein monofilament and/or a biothesiometer (Me.Te.Da. s.r.l., San Benedetto del Tronto AP, Italy) [21,22].

A senior Podiatrist (RJMB) performed all evaluations (demographic, vascular status, and neuropathic exploration).

The primary outcome measured in this study was the thickness of the sub-metatarsal fat pad in the first metatarsal.

A 7–12 mHz linear-array ultrasound transducer (MyLab 25 Gold model; Esaote SpA., Firenze, Italy) was used for assessing the soft tissue morphology. Participants were placed in a supine position with the ankle in a neutral position. Calluses were removed if necessary. A generous amount of ultrasound gel was placed between the skin and the transducer to avoid compressing the skin surface. The first and second metatarsals were measured to determine the plantar soft tissue thickness, as they are considered to be more reliable metatarsal areas and have lower statistical heterogeneity [23]. The ultrasound transducer was placed perpendicularly to the sesamoid bones and the second metatarsal head, and a longitudinal section of both the sub-metatarsal skin and the fat pad was obtained to measure the first and second metatarsal heads, respectively. First, metatarsal skin and fat pads were defined as the average of the measurements of the lateral and medial sesamoid [24]. The changes in the echogenicity of the ultrasound image on each sonogram were used to identify the soft tissue layers (Figure 1).

Two observers, who were each unaware of the measurements made by the other, took the measurements. The observers were podiatrists who had more than 3 years of experience in treating diabetic foot ulcers and were trained in soft-tissue ultrasound. In a unique session, each observer performed several trials to obtain two clear images, which were used to measure each area. The intra-reliability of the two measurements was calculated for each observer. The first measurement of each observer was used to calculate the inter-reliability. The intrasubject and intraclass correlation coefficients (ICCs) for both observers were almost perfectly in agreement (0.932–0.994 for observer A and 0.929–0.993 for observer B). The reliability of measurements between examiners, as demonstrated by ICCs ranging from 0.79 to 0.96 (Table 1), was considered substantial and in almost perfect agreement, respectively, according to the Landis and Koch classification [25].

The sample size was calculated using the Granmo v.12 program (Municipal Institute of Medical Research, Barcelona, Spain) (https://www.imim.cat/ofertadeserveis/software-public/granmo/). A previous study showed a fat pad thickness of 0.39 ± 0.11 cm in the first metatarsal head of patients with diabetes [26]. We considered 0.10 cm as relevant in identifying the differences between groups. Therefore, we analyzed 38 patients (19 in each group) with an alpha of 0.05 and a statistical power of 0.80.

### Statistical Analysis

The selected measurements of the first and second metatarsal head were calculated from the mean of the two measurements obtained by both clinicians for each area. In patients with a history of lateral or medial sesamoid resection, or whose second metatarsal head was removed, this area was excluded from the analysis.

Statistical analysis was performed using SPSS for IOs version 21.0 (SPSS, Inc. Chicago, IL, USA). The assumption of the normality of all continuous variables was verified using the Kolmogorov–Smirnov test. Statistical differences between groups were calculated using the Chi-Square test and, where appropriate, Fisher’s exact test for categorical variables. The Mann–Whitney U test was performed for abnormally distributed quantitative parameters, and Student’s *t*-test was performed for quantitative variables distributed normally. Differences between groups in the mean of skin and fat pad thickness were analyzed using Student’s *t*-test. The criteria of *p* < 0.05 was accepted as statistically significant with a confidence interval of 95%.

## 3. Results

Seven patients from the experimental group had a history of second metatarsal head resection, therefore only the data on the first metatarsal head were analyzed. Figure 2 shows the flow chart of the included patients. The third, fourth, and fifth metatarsals were resected in six, five, and five patients, respectively, from the experimental group. Metatarsal head resection was performed in the experimental group with a mean time of 6 ± 4.29 years. Moreover, five, six, six, three, and seven patients from the control group previously developed ulcers in the hallux, second, third, fourth, and fifth toes, respectively.

As expected, patients who underwent metatarsal head resection had a more severe SINBAD grade (3.1 ± 1.3 points) than patients in the control group (2.3 ± 7.5 points). No significant differences were observed in the other baseline characteristics between the two groups (Table 2).

The mean skin sub-metatarsal thickness of the sample was 2.64 ± 0.54 mm for the first metatarsal head and 2.47 ± 0.56 mm for the second metatarsal head. The mean sub-metatarsal fat pad thickness of the first metatarsal head was 4.59 ± 1.42 mm and that of the second metatarsal head was 3.88 ± 1.76 mm. The first or second sub-metatarsal skin thickness did not show significant differences between groups (*p* = 0.063, 95% CI: −0.019–0.672 and *p* = 0.066, CI: −0.027–0.786, respectively); however, both first and second sub-metatarsal fat pads were thinner in patients with previous metatarsal head resection than in the control group (*p* < 0.001, CI: 0.943–2.457 and *p* < 0.001, CI: 1.143–3.270, respectively) (Figure 3 and Figure 4).

## 4. Discussion

The results of this study show that patients with a history of metatarsal head resection have atrophy of the sub-metatarsal fat pad compared with those with previous ulceration in a different area of the metatarsal head. By contrast, no significant differences were observed in the sub-metatarsal skin thickness between patients with and without previous metatarsal head resection.

Sub-metatarsal fat pads are composed of adipocytes surrounded by firm fibrous tissue septa rich in elastin. These specific structures have shock absorption properties besides providing insulation and protection for the metatarsal heads [27]. Problems with the functions, especially in the state of atrophy, of these sub-metatarsal fat pads can increase peak plantar pressure in patients with diabetes [24]. According to our results, a metatarsal head with a thinner fat pad layer has less protective ability, possibly increasing the risk of ulceration and reulceration.

Fat pad tissue is exposed to structural and functional changes in different groups of patients, such as those with active ulcers [28] or digital deformities [29]. This suggests the high adaptability of fat pad tissue to inflammatory or high-pressure processes. We believe that the overload of plantar pressure in patients requiring metatarsal head resection causes the differences in the thickness of the sub-metatarsal fat pad compared with other groups of patients at high risk of neuropathic ulceration who did not develop metatarsal ulceration. This mechanical theory has been previously proposed in an attempt to change the paradigm which assumed that the decrease in plantar pressure is the result of hyperglycaemia and glycation end products [12].

Although plantar pressure decreases locally after metatarsal head resection [30], the rest of the metatarsals continue to support excessive loading; in fact, metatarsal head reulceration is a very common condition affecting more than 40% of the patients who underwent metatarsal head resection [17]. It has been described that structural changes and biomechanical alterations occur after the metatarsal head resection [31]. According to our results, those changes could be related to the sub-metatarsal fat pad. An accumulation of internal tissue damage could cause structural changes in the tissues intended for supporting the loads due to the high plantar pressure. The lack of statistical differences in the thickness of the sub-metatarsal skin between groups suggests that the plantar fat pad is responsible for managing the accumulative stress over time in the sub-metatarsal area, producing structural changes in the fat pad.

The relationship between aging and fat pad atrophy has also been studied. It has not been possible to demonstrate whether fat pad atrophy is a normal part of the aging process or whether it indicates pathology [32]. We have not found differences between age groups. Therefore, we deduce that pathology is more relevant than aging in plantar fat pad atrophy. Further research should address treatment(s) to reduce plantar pressure in patients with plantar fat pad atrophy, thereby decreasing the events of ulceration and reulceration and improving their quality of life [33].

The study has several limitations. Firstly, an observational study was used to explore the associations, but the casual relationships cannot be inferred from the results.

Secondly, seven patients with a history of metatarsal head resection should be excluded from the analysis due to the conduct of metatarsal head resection. However, the first metatarsal was evaluated in the entire sample; this is the most reliable area for reference measurement of the plantar soft tissues [23]. Analysis of the second metatarsal could be deleted from the study due to the possible risk of statistical errors caused by the loss of data. The differences in the sub-metatarsal fat pad of the first metatarsal could be underestimated due to the individual characteristics of this metatarsal. Statistical differences in the second metatarsal heads compared to those of the first metatarsal heads primarily represent decreasing sub-metatarsal fat pad atrophy.

Alterations in skin hydration are common among patients with diabetes [34]. Less hydration has been associated with thinner skin and measurements of skin, derma, and fat can be affected by different hydration statuses [35]. We did not evaluate skin hydration in this study; however, it could modify the results.

Finally, our results could contribute to an understanding of the evolutionary process of the atrophy of plantar fat pads to inform the implementation of appropriate preventions or treatment measures.

## 5. Conclusions

This is the first study to evaluate the thickness of sub-metatarsal skin and fat pads in two groups of neuropathic patients with diabetes at a higher risk of ulceration. The results showed that atrophy of the sub-metatarsal fat pad is a differential factor in patients with history of metatarsal head resection, which could be associated with the risk of deep ulceration and transfer lesion.

## Figures and Tables

**Figure 1 jcm-09-00794-f001:**
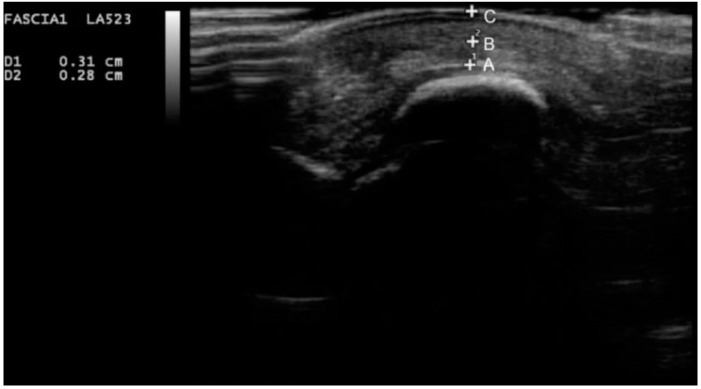
Ultrasound image of the sub-metatarsal skin and fat pad thickness under the medial sesamoid. Point A to point B, representing the thickness of the sub-metatarsal fat pad; point B to point C, representing the thickness of the sub-metatarsal skin.

**Figure 2 jcm-09-00794-f002:**
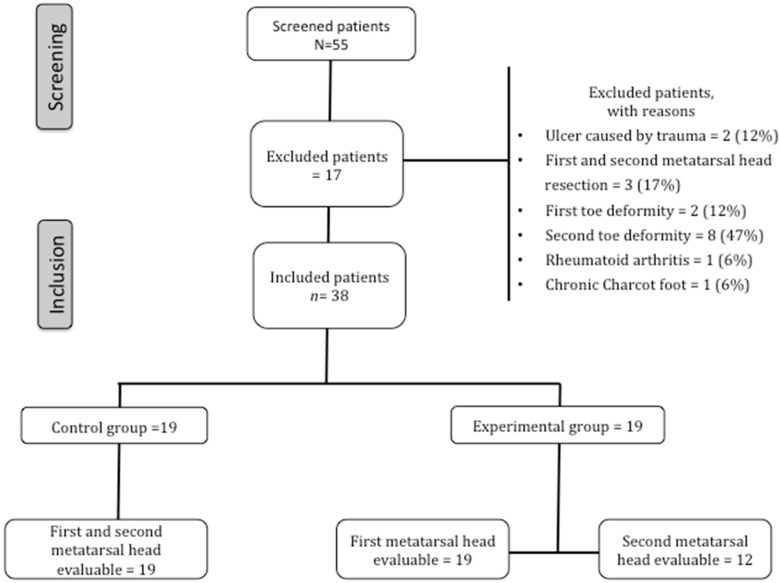
Flow chart of the included patients.

**Figure 3 jcm-09-00794-f003:**
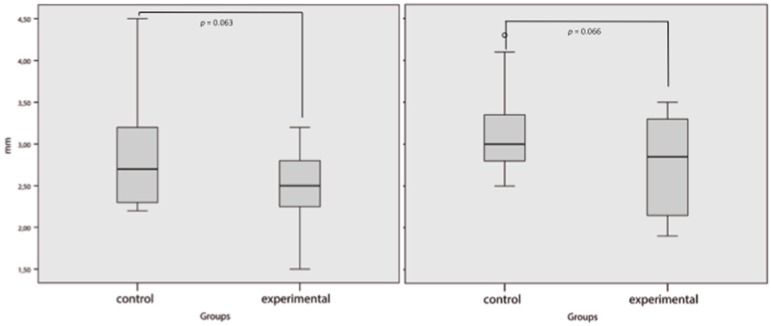
Sub-metatarsal skin thickness. Left figure for first metatarsal head and right figure for second metatarsal head. *p* value < 0.05 was considered significant.

**Figure 4 jcm-09-00794-f004:**
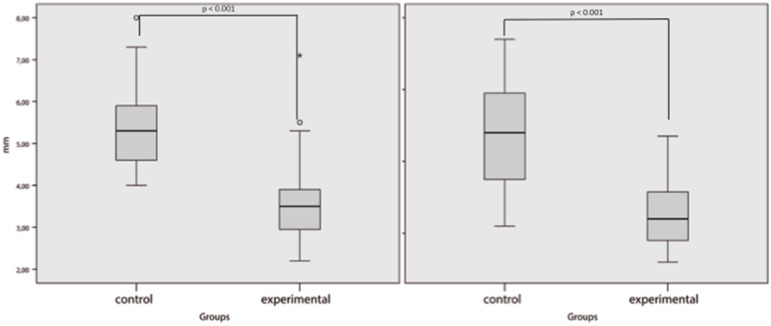
Sub-metatarsal fat pad. Left figure for first metatarsal and right figure for second metatarsal. *p* value < 0.05 was considered significant.

**Table 1 jcm-09-00794-t001:** Intra-subject and inter-subject reliability measurements.

*n* = 38	Intra-Subject Reliability ICC (95% CI)		Inter-Subject Reliability ICC (95% CI)
	Observer A	Observer B	
1st Metatarsal Plantar Skin	958 * (919–978)	959 * (921–979)	795 * (606–894)
2nd Metatarsal Plantar Skin	932 * (855–968)	929 * (849–967)	779 * (529–896)
1st Metatarsal Fat Pad	993 * (986–996)	993 * (986–996)	962 * (926–980)
2nd Metatarsal Fat Pad	994 * (988–997)	989 * (977–995)	952 * (898–978)

Abbreviations: ICCs = intraclass correlation coefficients, CI = confidence interval. * *p* value < 0.05 was considered significant.

**Table 2 jcm-09-00794-t002:** Baseline data of the sample.

(*n* = 38 Patients)	Control Group *n* = 19	Experimental Group *n* = 19	*p*-Value
Male/Female *n* (%)	14 (74)/5 (26)	13 (68)/6 (32)	0.721
Type 1/Type 2 DM *n* (%)	4 (21)/15 (79)	1 (5)/18 (95)	0.150
PAD *n* (%)	12 (63)	10 (53)	0.511
Retinopathy *n* (%)	4 (21)	5 (26)	0.703
Nephropathy *n* (%)	3 (16)	2 (11)	0.631
Mean age ± SD (Years)	60 ± 11.1	65 ± 9.4	0.124
Diabetes Mellitus (Years), Mean ± SD	23 ± 18.4	21 ± 12.9	0.746
SINBAD Classification Score (Points), Mean ± SD	2.3 ± 7.5	3.1 ± 1.3	0.036
Glycated Haemoglobin mmol/mol, Mean ± SD	53.57 ± 8.83	58.61 ± 16.63	0.254
Body Mass Index (kg/cm^2^), Mean ± SD	29.84 ± 5.91	28.69 ± 5.47	0.537

Abbreviations: DM = diabetes mellitus. *p* value < 0.05 was considered significant.

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
