# Peer review of "Differences in the Sub-Metatarsal Fat Pad Atrophy Symptoms between Patients with Metatarsal Head Resection and Those without Metatarsal Head Resection: A Cross-Sectional Study"

_jcm, 2020, doi:10.3390/jcm9030794_

Round 1

Reviewer 1 Report

The importance of skin and dermal hydration has been discussed in different paper in the past. Mesuerements of skin, derma and fat can be affected by different hydration status. If the author considered this status during the study, they should use in the paper. Otherwise  Author scould consider to add  this aspect, at least in the  discussion

Author Response

Reviewer 1

Comments and Suggestions for Authors

The importance of skin and dermal hydration has been discussed in different paper in the past. Measurements of skin, derma and fat can be affected by different hydration status. If the author considered this status during the study, they should use in the paper. Otherwise, author should consider to add  this aspect, at least in the  discussion

Thank you for accepting the review of the current article and for the suggestions made, which could improve considerably the quality of the manuscript. Here, below your comments, you can see the comments and changes made after revision in highlight.

We have discussed the relevancy of skin hydration and we have identified it as a limitation of the study according to the recommendations

Reviewer 2 Report

This study compared plantar skin and fat pad thickness in two groups of patients, 1) patients with a heal plantar MT ulcer which required MT head resection (experimental group) and 2) patients with a healed plantar MT ulcer which did not undergo MT resection (control groups). All patients in this study had diabetes mellites. Overall, this is generally a well written manuscript and the described study adds to deficiencies within the current literature, although there are number of important issues which require attention.

Major changes:

A major concern is how comparable both study groups are. The experimental group is formed of patients who required MT head resection to heal their ulcers, whilst the control group did not require bony resection. This obviously suggests that patients in the control group had more severe ulceration (e.g. deep tissue infection/osteomyelitis) and therefore potentially more likely to have resultant skin/fat pad atrophy. Further information (e.g. SINBAD score or Society of Vascular Surgery WIfI grades) is required to help better define these groups, otherwise it is difficult to draw conclusions for these data.

Other changes:

Abstract:

Line 18 – Please specify that both groups had no active ulceration.

Introduction:

Line 52: It should also be stated that MT head resection is important in the treatment of osteomyelitis

Line 54: It should be stated that MT head resection leads to changes in foot biomechanics which contributes to ulceration.

Methods:

Line 64: It is stated that patients were recruited from those admission to the diabetic foot unit, however all patients had a history of healed ulceration. Were these patients admitted for other diabetic foot complication? A line specifying the reasons for admission would be useful here.

Line 69: Did all patients in the control group have previous ulceration over the plantar surface of the first/second MT. If not, is comparison of first/second MT skin/fat pad really valid in these patients? Please clarify.

Line 77: Please change ‘distal’ to ‘pedal’.

Line 78: The definition of PAD runs the risk of underdiagnosing the condition, as both pedal pulses had to be absent +/- the ABI was required to be <0.9. ABIs in this cohort of patients is unreliable and therefore patients with only a single pedal pulse and falsely normal ABI could be labelled as not having the condition. Can you specify how many patients fit into this category and what has been done to minimise this?

Line 77- 81 – Please specify who performed the assessment of peripheral vasculature and neuropathy.

Line 86 -  Please specify who performed the ultrasound analysis and whether they are trained in performing soft tissue ultrasound.

Line 108 – Based upon the Landis and Koch classification, the k statistics range between ‘substantial’ to ‘almost perfect’ agreement. Please correct with the correct terminology.

Line 115 – Please specify why 0.10cm is deemed relevant for identifying difference between groups (is this based upon the SD identified in reference 22?). Does this difference hold for skin thickness?

Line 123-4 - The description of statistical methods seems incorrect and needs revision. 1) The study compared mean skin and fat pad thicknesses between groups. It did not investigate for factors associated with changes in these thicknesses. This statement should be revised. 2) All data are assumed as being parametric for the basis of statistical testing, however this may not be correct. Examination via histograms/Komogorov-Sminov test is required and handling adjusted accordingly. 3) The statistical tests described as incorrect for this analysis. Categorical variables should be tested using the x2/Fisher’s exact test and continuous variables through either a student’s t/Mann-Whitney U test, depending on the distribution of the data. 4) More detailed is required on how data will be presents (e.g. confidence intervals) and the level set for statistical significance.  

Results:

A study flow diagram would be a useful addition showing numbers screened, included and excluded (with reasons).

Line 128 – Seven patients (37%) in the experimental groups were excluded from the analysis of the second MT soft tissues. This analysis is therefore underpowered and at risk of statistical error. I would suggest the authors considered removing this from the analysis as the results may not be valid.

Table 2 – Please present glycated haemoglobin using mmol/mol, not %

Line 138-140 – Please present the full results of the analysis (as done in the abstract), including confident intervals.

Figure 2 and 3 – The image quality is poor and the data labels are very small and difficult to read. Please adjust accordingly.

Discussion:

Line 152-169 – These three paragraphs quote studies which have little relevance to the study soft tissue thickness and MT head resection (e.g. comparison of ulceration vs non-ulcerated feet) and actually make the discussion section quite confusing to the reader. If these studies are to be quoted, then they should be written in terms of their application to the current study.

Line 175 – Please specify that fat pad atrophy ‘may’ increase the risk of ulceration/reulceration. Difference in observed soft tissue thickness, not risk of ulceration has not been studied in this current study.

Line 194 – As per previous point – if patients were excluded then can analysis of the 2ndMT soft tissue be undertaken. If it is to be included, this should be justified in more detail here.

Conclusion:

No issues

Author Response

Reviewer 2

Comments and Suggestions for Authors

This study compared plantar skin and fat pad thickness in two groups of patients, 1) patients with a heal plantar MT ulcer which required MT head resection (experimental group) and 2) patients with a healed plantar MT ulcer which did not undergo MT resection (control groups). All patients in this study had diabetes mellites. Overall, this is generally a well written manuscript and the described study adds to deficiencies within the current literature, although there are number of important issues which require attention.

Thank you for accepting the review of the current article and for the suggestions made, which could improve considerably the quality of the manuscript. Here, below your comments, you can see the comments and changes made after revision in highlight.

Major changes:

A major concern is how comparable both study groups are. The experimental group is formed of patients who required MT head resection to heal their ulcers, whilst the control group did not require bony resection. This obviously suggests that patients in the control group had more severe ulceration (e.g. deep tissue infection/osteomyelitis) and therefore potentially more likely to have resultant skin/fat pad atrophy. Further information (e.g. SINBAD score or Society of Vascular Surgery WIfI grades) is required to help better define these groups, otherwise it is difficult to draw conclusions for these data.

We think that WIFI classification does not apply, because is recently recommended to stratify amputation risk and revascularization benefit in a patient with a diabetic foot ulcer and peripheral artery disease, and our patients did not have active ulcer at the inclusion of the study:

 https://iwgdfguidelines.org/wp-content/uploads/2019/05/04-IWGDF-PAD-guideline-2019.pdf

We have performed a Sinbad classification system and the results have been shown in the table 2.  We have added this issue in the methods and results sections. As the reviewer point, patients in the experimental group had deeper and more severe ulcers. We have not considered to discuss this concern because the soft tissues of the specific metatarsal head resected were excluded of the analysis by ultrasound.

Other changes:

Abstract:

 Line 18 – Please specify that both groups had no active ulceration.

This has been changed in the text as suggested.

Introduction:

Line 52: It should also be stated that MT head resection is important in the treatment of osteomyelitis

This has been changed in the text according to the recommendation.

Line 54: It should be stated that MT head resection leads to changes in foot biomechanics which contributes to ulceration.

It was added in the introduction section as suggested.

Methods:

Line 64: It is stated that patients were recruited from those admission to the diabetic foot unit, however all patients had a history of healed ulceration. Were these patients admitted for other diabetic foot complication? A line specifying the reasons for admission would be useful here.

It was added in the methods section according to the recommendation.

Line 69: Did all patients in the control group have previous ulceration over the plantar surface of the first/second MT. If not, is comparison of first/second MT skin/fat pad really valid in these patients? Please clarify.

None patient had history of ulcer in the first or second metatarsal head in the control group. Control group was defined as, patients with history of DFU located in a different location than first, second, third, fourth or five metatarsal.

Finally, all the patients included in the control group had history of ulcer in the toes. Our objective was to evaluate the differences between the skin/fat pad thickness when these tissues were undamaged. A previous ulcer beneath the first or second metatarsal should break these structures and distort the results of the ultrasound thickness. 

Line 77: Please change ‘distal’ to ‘pedal’.

This word has been changed in the text, accordingly.

Line 78: The definition of PAD runs the risk of underdiagnosing the condition, as both pedal pulses had to be absent +/- the ABI was required to be <0.9. ABIs in this cohort of patients is unreliable and therefore patients with only a single pedal pulse and falsely normal ABI could be labelled as not having the condition. Can you specify how many patients fit into this category and what has been done to minimise this?

We have changed the definition of peripheral artery disease. We have re-calculated the PAS according to data of TBI in patients with ABI > 0.9 as suggested. Changes have been made in the table 2 and methods section in consequence.

Line 77- 81 – Please specify who performed the assessment of peripheral vasculature and neuropathy.

A senior podiatric (RJMB) performed all evaluations (demographic, vascular status, and neuropathic exploration). This has been added in the text according the recommendations.

Line 86 -  Please specify who performed the ultrasound analysis and whether they are trained in performing soft tissue ultrasound.

The observer who performed the ultrasound exploration was described in the text. We have added the experience in the training with ultrasound in the text as suggested “The observers were podiatrists who had more than 3 years of experience in treating diabetic foot ulcers and were trained in soft-tissue ultrasound”

Line 108 – Based upon the Landis and Koch classification, the k statistics range between ‘substantial’ to ‘almost perfect’ agreement. Please correct with the correct terminology.

Terminology has been modified accordingly.

Line 115 – Please specify why 0.10cm is deemed relevant for identifying difference between groups (is this based upon the SD identified in reference 22?). Does this difference hold for skin thickness?

Sample size calculation was based on the differences between the first metatarsal fat pads on the reference 22. We changed the primary outcome to fat pad thickness, only (excluding the skin thickness), in order to calculate the sample size according to recommendation.

 Line 123-4 - The description of statistical methods seems incorrect and needs revision. 1) The study compared mean skin and fat pad thicknesses between groups. It did not investigate for factors associated with changes in these thicknesses. This statement should be revised. 2) All data are assumed as being parametric for the basis of statistical testing, however this may not be correct. Examination via histograms/Komogorov-Sminov test is required and handling adjusted accordingly. 3) The statistical tests described as incorrect for this analysis. Categorical variables should be tested using the x2/Fisher’s exact test and continuous variables through either a student’s t/Mann-Whitney U test, depending on the distribution of the data. 4) More detailed is required on how data will be presents (e.g. confidence intervals) and the level set for statistical significance. 

We have modified the description of the statistical methods according to the recommendation:  1) the text; 2) Kolmogorov/Smirnov 3) the tests and 4) The level set.

Results:

A study flow diagram would be a useful addition showing numbers screened, included and excluded (with reasons).

We have added a flow chart as suggested.

Line 128 – Seven patients (37%) in the experimental groups were excluded from the analysis of the second MT soft tissues. This analysis is therefore underpowered and at risk of statistical error. I would suggest the authors considered removing this from the analysis as the results may not be valid.

We agree with this comment, however the aim of the study is to evaluate the metatarsal fat pad. If we evaluate one metatarsal, it is exposed a local consideration for the readers and can lose the initial intention of the study. We prefer maintain this analysis, in spite of risk of statistical error, and consider it as a limitation of the study.

Table 2 – Please present glycated haemoglobin using mmol/mol, not %

Glycated haemoglobin has been expressed as mmol/mol, as suggested

Line 138-140 – Please present the full results of the analysis (as done in the abstract), including confident intervals.

Confident intervals have been included in the results section as recommended.

Figure 2 and 3 – The image quality is poor and the data labels are very small and difficult to read. Please adjust accordingly.

Fig. 2 and 3 (currently 3 and 4 after changes) have been modified according the recommendations.

Discussion:

Line 152-169 – These three paragraphs quote studies which have little relevance to the study soft tissue thickness and MT head resection (e.g. comparison of ulceration vs non-ulcerated feet) and actually make the discussion section quite confusing to the reader. If these studies are to be quoted, then they should be written in terms of their application to the current study.

These paragraphs have been deleted and the applicable information of them has been summarized in a new paragraph as suggested. 

Line 175 – Please specify that fat pad atrophy ‘may’ increase the risk of ulceration/reulceration. Difference in observed soft tissue thickness, not risk of ulceration has not been studied in this current study.

This issue has been modified as suggested.

Line 194 – As per previous point – if patients were excluded then can analysis of the 2ndMT soft tissue be undertaken. If it is to be included, this should be justified in more detail here.

This issue has been widely justified according to previous comments.

Conclusion:

No issues

Reviewer 3 Report

I am grateful for the possibility to revise this research study.

Differences in the Sub-Metatarsal Fat Pad Atrophy Symptoms Between Patients with Metatarsal Head Resection and Those Without Metatarsal Head  Resection: A Cross-Sectional Study is a trend topic in the current research literature and may be a main focus of interest for readers.

Results of the abstract need to reflect the findings with respect to both groups and the lack of significant differences for pronation

Introduction may be improved adding new information in order to provide an adequate state-of-the-art. Furthermore, a hypothesis is lacking.

I also suggest some change in grammar.In line 36 change the word "occur" by the word "appear"

Methods are well-designed with relevant and complete information. Correct sample size calculations, good description of the properties of the outcome measurements as well as detailed statistical analyses were included. I suggest to include this references include in the atteched to complet this requeriment

Chicharro-Luna, E.; Pomares-Gómez, F.J.; Ortega-Ávila, A.B.; Marchena-Rodríguez, A.; Blanquer-Gregori, J.F.J.; Navarro-Flores, E. Predictive model to identify the risk of losing protective sensibility of the foot in patients with diabetes mellitus. Int. Wound J. 2020, 17, 220–227.

Tables, figures and redaction of the results are presented in a correct way providing a good presentation of the main finding of the study. I suggest to include this references include in the atteched to complet this requeriment

López-López, D.; Becerro-de-Bengoa-Vallejo, R.; Losa-Iglesias, M.E.; Soriano-Medrano, A.; Palomo-López, P.; Morales-Ponce, Á.; Rodríguez-Sanz, D.; Calvo-Lobo, C. Relationship between decreased subcalcaneal fat pad thickness and plantar heel pain. a case control study. Pain Physician 2019, 22, 109–116.

Discussion may include future research studies secondary to the current findings of this study. Clinical considerations, limitations and overall discussion are well-presented, but future research may be useful in order to propose future research regarding this field.

Furthermore, discuss the possible influence of aging in your study finding suggest to include this references include in the atteched to complet this requeriment

Navarro-Flores, E.; Pérez-Ros, P.; FM, M.-A.; Julían-Rochina, I.; Cauli, O. Neuro-psychiatric alterations in patients with diabetic foot syndrome. CNS Neurol. Disord. - Drug Targets 2019, 18.

Navarro-Flores, E.; Cauli, O. Quality of life in individuals with diabetic foot syndrome. Endocrine, Metab. Immune Disord. - Drug Targets 2020, 20.

Author Response

Reviewer 3

Comments and Suggestions for Authors

I am grateful for the possibility to revise this research study.

Differences in the Sub-Metatarsal Fat Pad Atrophy Symptoms Between Patients with Metatarsal Head Resection and Those Without Metatarsal Head  Resection: A Cross-Sectional Study is a trend topic in the current research literature and may be a main focus of interest for readers.

Thank you for accepting the review of the current article and for the suggestions made, which could improve considerably the quality of the manuscript. Here, below your comments, you can see the comments and changes made after revision in highlight.

Results of the abstract need to reflect the findings with respect to both groups and the lack of significant differences for pronation

We are not completely convinced of understanding the recommendation. We have not evaluated pronation (as foot type) in this study and thus we can not show it in the results section. We have shown the statistical differences that it was found between groups in the variables evaluated (fat pad and skin). Please, clarify if you consider it necessary.

Introduction may be improved adding new information in order to provide an adequate state-of-the-art. Furthermore, a hypothesis is lacking.

New information and hypothesis have been added in the introduction section accordingly.

I also suggest some change in grammar.In line 36 change the word "occur" by the word "appear"

We have modified the text as suggested and the manuscript has been sent to language edition again.

Methods are well-designed with relevant and complete information. Correct sample size calculations, good description of the properties of the outcome measurements as well as detailed statistical analyses were included. I suggest to include this references include in the atteched to complet this requeriment

Chicharro-Luna, E.; Pomares-Gómez, F.J.; Ortega-Ávila, A.B.; Marchena-Rodríguez, A.; Blanquer-Gregori, J.F.J.; Navarro-Flores, E. Predictive model to identify the risk of losing protective sensibility of the foot in patients with diabetes mellitus. Int. Wound J. 2020, 17, 220–227.

We have added the reference according to the recommendation.

Tables, figures and redaction of the results are presented in a correct way providing a good presentation of the main finding of the study. I suggest to include this references include in the atteched to complet this requeriment

López-López, D.; Becerro-de-Bengoa-Vallejo, R.; Losa-Iglesias, M.E.; Soriano-Medrano, A.; Palomo-López, P.; Morales-Ponce, Á.; Rodríguez-Sanz, D.; Calvo-Lobo, C. Relationship between decreased subcalcaneal fat pad thickness and plantar heel pain. a case control study. Pain Physician 2019, 22, 109–116.

 We have added the reference according to the recommendation.

Discussion may include future research studies secondary to the current findings of this study. Clinical considerations, limitations and overall discussion are well-presented, but future research may be useful in order to propose future research regarding this field.

Furthermore, discuss the possible influence of aging in your study finding suggest to include this references include in the atteched to complet this requeriment

Aging has been discussed as suggested and the reference has been added as suggested.

Navarro-Flores, E.; Pérez-Ros, P.; FM, M.-A.; Julían-Rochina, I.; Cauli, O. Neuro-psychiatric alterations in patients with diabetic foot syndrome. CNS Neurol. Disord. - Drug Targets 2019, 18.

Navarro-Flores, E.; Cauli, O. Quality of life in individuals with diabetic foot syndrome. Endocrine, Metab. Immune Disord. - Drug Targets 2020, 20.

Round 2

Reviewer 2 Report

Thank you for this opportunity to re-review this manuscript. The authors have carefully considered and answer all the points raised. I have no further comments to add.